# Genetic and Chemical Controls of Sperm Fate and Spermatocyte Dedifferentiation via PUF-8 and MPK-1 in *Caenorhabditis elegans*

**DOI:** 10.3390/cells12030434

**Published:** 2023-01-28

**Authors:** Youngyong Park, Matthew Gaddy, Moonjung Hyun, Mariah E. Jones, Hafiz M. Aslam, Myon Hee Lee

**Affiliations:** 1Division of Hematology/Oncology, Department of Internal Medicine, Brody School of Medicine at East Carolina University, Greenville, NC 27834, USA; 2Biological Resources Research Group, Bioenvironmental Science & Toxicology Division, Korea Institute of Toxicology, Jinju 52834, Gyeongsangnam-do, Republic of Korea; 3Department of Biology, East Carolina University, Greenville, NC 27858, USA

**Keywords:** PUF-8, MPK-1, sperm fate, dedifferentiation, resveratrol, *C. elegans* germline

## Abstract

Using the nematode *C. elegans* germline as a model system, we previously reported that PUF-8 (a PUF RNA-binding protein) and LIP-1 (a dual-specificity phosphatase) repress sperm fate at 20 °C and the dedifferentiation of spermatocytes into mitotic cells (termed “spermatocyte dedifferentiation”) at 25 °C. Thus, double mutants lacking both PUF-8 and LIP-1 produce excess sperm at 20 °C, and their spermatocytes return to mitotically dividing cells via dedifferentiation at 25 °C, resulting in germline tumors. To gain insight into the molecular competence for spermatocyte dedifferentiation, we compared the germline phenotypes of three mutant strains that produce excess sperm—*fem-3(q20gf)*, *puf-8(q725)*; *fem-3(q20gf*), and *puf-8(q725)*; *lip-1(zh15)*. Spermatocyte dedifferentiation was not observed in *fem-3(q20gf)* mutants, but it was more severe in *puf-8(q725)*; *lip-1(zh15)* than in *puf-8(q725)*; *fem-3(q20gf)* mutants. These results suggest that MPK-1 (the *C. elegans* ERK1/2 MAPK ortholog) activation in the absence of PUF-8 is required to promote spermatocyte dedifferentiation. This idea was confirmed using Resveratrol (RSV), a potential activator of MPK-1 and ERK1/2 in *C. elegans* and human cells, respectively. Notably, spermatocyte dedifferentiation was significantly enhanced by RSV treatment in the absence of PUF-8, and its effect was blocked by *mpk-1* RNAi. We, therefore, conclude that PUF-8 and MPK-1 are essential regulators for spermatocyte dedifferentiation and tumorigenesis. Since these regulators are broadly conserved, we suggest that similar regulatory circuitry may control cellular dedifferentiation and tumorigenesis in other organisms, including humans.

## 1. Introduction

The nematode *C. elegans* is a multicellular organism that has become a popular model for biological and basic medical research. It has also been widely used as a model system to explore fundamental questions in multiple aspects of biology, including development, stem cell regulation, cell fate decision, tumorigenesis, and aging [1,2,3,4,5]. *C. elegans* has two sexes: male and hermaphrodite. In males and hermaphrodites, spermatogenesis begins in the L4 larval stage [6]. Spermatogenesis continues throughout the lifetime of a male, whereas it ceases and switches to oogenesis in the late L4 stage in a hermaphrodite [6] (Figure 1A).

Development of the *C. elegans* germline progresses by many of the same steps typical of other animal germlines [8]. The *C. elegans* germline is organized in a simple linear fashion that progresses from germline stem cells (GSCs) at one end to maturing gametes at the other (Figure 1A). Germ cells progress from GSCs at the distal end through meiotic prophase as they move proximally to become differentiated gametes at the proximal end [6]. *C. elegans* germline development is tightly regulated by conserved external signaling pathways, including GLP-1/Notch signaling, and intrinsic regulators, including RNA binding proteins and cell cycle regulators [9] (Figure 1B). The Notch signaling pathway and its core components in *C. elegans* are highly conserved. *C. elegans* has two Notch receptors, GLP-1 and LIN-12, which mediate cell-cell interaction during the development [10]. Specifically, GLP-1/Notch signaling in the *C. elegans* germline is critical for GSC maintenance and continued mitotic division through its direct target genes—*sygl-1* and *lst-1* [9,11] (Figure 1B). In addition to GLP-1/Notch signal pathways, a battery of RNA regulators, including PUF (Pumilio/FBF) RNA-binding proteins, play critical roles in GSC maintenance, differentiation, and cell fate specification in the *C. elegans* germline [7] (Figure 1B). In vertebrates, PUF proteins control various physiological processes such as stem cell proliferation [12,13], tumorigenesis [13], neurogenesis [14,15], germline development [16,17], mesenchymal cell fate decision [12], and mitochondrial dynamics/mitophagy [18] by interacting with the 3′ untranslated regions (UTRs) of specific mRNAs to repress the mRNA translation or stability (Figure 1C).

*C. elegans* has 11 PUF proteins that recognize a family of related sequence motifs in the target mRNAs (Figure 1D), yet individual PUF proteins have distinct biological functions [19]. Among them, the PUF-8 (mainly like the *Drosophila* and human PUFs) protein controls multiple cellular processes, including GSC proliferation, differentiation, dedifferentiation, and sperm-oocyte decision, depending on the genetic context [20] (Figure 1E). Most *puf-8(q725* or *ok302)* single mutants make both sperm and oocytes, and they are self-fertile at the permissive temperature (20 °C) [21,22,23,24]. However, MPK-1 activation promotes sperm fate, resulting in masculinization of the germline (Mog) phenotype at 20 °C [25,26] and spermatocyte dedifferentiation, resulting in germline tumors at 25 °C in the absence of PUF-8 [21,23] (Figure 1E,F). Dedifferentiation is a cellular process by which cells from partially or terminally differentiated stages revert to a less differentiated stage. This cellular phenomenon has been implicated in regenerative medicine and tumorigenesis [27]. Although this cellular process is observed in vivo in many eukaryotes, its cellular mechanism remains poorly understood.

To better understand the mechanism of sperm fate specification and spermatocyte dedifferentiation, we have generated a *puf-8(q725)*; *fem-3(q20gf)* double mutant. The *puf-8(q725)*; *fem-3(q20gf)* mutants exhibit a similar germline phenotype (Mog phenotype) as seen in *puf-8(q725)*; *lip-1(zh15)* mutant germlines at 20 °C, but they had significantly less dedifferentiation-mediated germline tumors than *puf-8(q725)*; *lip-1(zh15)* at 25 °C. Our genetic and chemical analyses demonstrated that sperm fate and spermatocyte dedifferentiation require PUF-8 loss and MPK-1 activation in the *C. elegans* germline. Activation of MPK-1 by resveratrol treatment in the absence of PUF-8 significantly promotes sperm fate and spermatocyte dedifferentiation. Mammalian ERK1/2 MAPK has also been implicated in cellular dedifferentiation [28,29,30,31,32]. Therefore, our findings provide insights into cellular dedifferentiation and tumorigenesis in other organisms, including humans.

## 2. Materials and Methods

### 2.1. Worm Maintenance and Strains

*C. elegans* strains were maintained at 20 °C or 25 °C as previously described [1]. *C. elegans* strains were provided by Caenorhabditis Genetics Center (CGC) and Dr. Kimble’s lab (University of Wisconsin-Madison, Madison, WI, USA) or generated by us using a standard genetic method. Appendix A lists strains used in this study.

### 2.2. RNA Interference (RNAi)

RNAi experiments were performed by feeding bacteria expressing double-strand RNAs (dsRNAs) corresponding to the gene of interest [33]. Briefly, synchronized L1 staged worms were plated onto RNAi plates and incubated at 25 °C. Germline phenotypes were determined by staining dissected gonads with specific markers and DAPI. For *mpk-1b* RNAi, the unique region (exon 1; 1–240 nt) of the *mpk-1b* gene was amplified by PCR from *C. elegans* genomic DNA and cloned into the pPD129.36 (L4440) vector containing two convergent T7 polymerase promoters in opposite orientations separated by a multi-cloning site [25,34]. Other RNAi bacteria were from the *C. elegans* RNAi feeding library (Source Bioscience LifeSciences) and *C. elegans* ORF-RNAi library (Open Biosystems).

### 2.3. Generation of puf-8(q725)/mln1[mIs14 dpy-10(e128)]; fem-3(q20gf) Mutants

Three adult *puf-8(q725)* homozygote male mutants were mated with five adult *fem-3(q20)* gain-of-function (gf) homozygote hermaphrodite mutants at 20 °C. Male progeny (predicted genotype: *puf-8(q725)/+*; *fem-3(q20gf)/+*) were mated with dumpy homozygote hermaphrodite mutants with an *LGII mln1[mIs14 dpy-10(e128)]* GFP balancer chromosome. Non-dumpy, pharyngeal GFP-positive hermaphrodites were selected and singled out. *puf-8(q725)/mln1[mIs14 dpy-10(e128)]*; *fem-3(q20gf)/+* progeny were identified by PCR and phenotype analysis at 25 °C (~25% F1 progeny exhibited Mog phenotype). Finally, *puf-8(q725)/mln1[mIs14 dpy-10(e128)]*; *fem-3(q20gf)/fem-3(q20gf)* double hermaphrodite mutants were identified by PCR and phenotype analysis in the next generation.

### 2.4. Generation of lin-41(tn1541[GFP::tev::s::lin-41]); puf-8(q725)/mln1[mIs14 dpy-10(e128)]; fem-3(q20gf) Mutant

Three adult *puf-8(q725)/mln1[mIs14 dpy-10(e128)]; fem-3(q20gf)* homozygote male mutants were mated with five adult *lin-41(tn1541[GFP::tev::s::lin-41])* homozygote hermaphrodites at 20 °C. Male progeny (predicted genotype: *lin-41(tn1541[GFP::tev::s::lin-41])/+; puf-8(q725)/+*; *fem-3(q20gf)/+*) were mated with dumpy homozygote hermaphrodite mutants with an *LGII mln1[mIs14 dpy-10(e128)]* GFP balancer chromosome. Next, non-dumpy, pharyngeal GFP-positive hermaphrodites were selected and singled out. *lin-41(tn1541[GFP::tev::s::lin-41])/+*; *puf-8(q725)/mln1[mIs14 dpy-10(e128)]*; *fem-3(q20gf)/+* progeny were identified by PCR, oocyte GFP expression, and phenotype analysis at 25 °C (~25% F1 progeny exhibited Mog phenotype). Finally, *lin-41(tn1541[GFP::tev::s::lin-41])/lin-41(tn1541[GFP::tev::s::lin-41])*; *puf-8(q725)/mln1[mIs14 dpy-10(e128)]*; *fem-3(q20gf)/fem-3(q20gf)* hermaphrodite mutants were identified by PCR, oocyte GFP expression, and phenotype analysis in the next generation.

### 2.5. Germline Antibody Staining

For germline antibody staining, dissected gonads were fixed in 4% paraformaldehyde/1xPBS solution (VWR, Radnor, PA, #AAJ61899-AK) for 20 min at 25 °C followed by 100% cold methanol for 5 min at −20 °C as described in [35]. After blocking for 30 min with 0.5% Bovine Serum Albumin (BSA, Sigma-Aldrich, St. Louis, MO, USA, #A7030) in 1 × PBST (1 × PBS + 0.1% Tween 20), fixed gonads were incubated overnight at 4 °C with primary antibodies followed by 1 h at 25 °C with secondary antibodies. Appendix A lists antibodies used in this study.

### 2.6. 5-Ethynyl-2′-deoxyuridine (EdU) Labeling

To label mitotically cycling cells, worms were incubated with rocking in 0.2 mL M9 buffer (3 g KH_2_PO_4_, 6 g Na_2_HPO_4_, 5 g NaCl, 1 mL 1M MgSO_4_, H_2_O to 1 L) containing 0.1% Tween 20 and 1 mM EdU for 30 min at 20 °C. Gonads were dissected and fixed in 3% paraformaldehyde/0.1M K_2_HPO_4_ (pH 7.2) solution for 20 min, followed by −20 °C methanol fixation for 10 min. Fixed gonads were blocked in 1 × PBST/0.5% BSA solution for 30 min at 20 °C. EdU labeling was performed using the Click-iT EdU Alexa Fluor 488 Imaging Kit (Thermo Fisher Scientific, Waltham, MA, USA, #C10337), according to the manufacturer’s instructions. For co-staining with antibodies, EdU-labeled gonads were incubated in the primary antibodies after washing three times and subsequently in the secondary antibodies as described above.

### 2.7. Resveratrol (RSV) Treatment

RSV (Sigma-Aldrich, St. Louis, MO, USA; Cat# R5010) was dissolved in ethanol (EtOH) to stock concentrations of 100 mM. RSV was directly added to the NGM media before pouring the solution into Petri dishes. The worms were transferred to the EtOH- or RSV-containing NGM agar plates. All worms tested were transferred to fresh plates every 2 days, and their germline phenotypes were determined by staining dissected gonads with cell type-specific antibodies or an EdU-labeling kit.

### 2.8. Cell Culture and RSV Treatment

MDA MB231 and WPMY-1 cells were grown in an appropriate growth culture medium (Dulbecco’s Modified Eagle’s Medium (DMEM) with sodium pyruvate) supplemented with 10% FBS (Thermo Fisher Scientific, Waltham, MA, USA, #10082147) and penicillin/streptomycin (10,000 U/mL). Cells were incubated with RSV for 24h after 80% confluence.

### 2.9. Western Blot Analysis

Cells were lysed as previously described [36]. Proteins were subjected to 10% SDS PAGE. Gels were transferred to the iblot transfer stack (Invitrogen, MA, USA, #IB4010 01) using a transfer apparatus (Invitrogen iBlot 2). Primary antibody incubations were performed in a blocking solution (5% BSA, 1 × TBS, 0.1% Tween20) overnight at 4 °C after blocking for 1 h in the blocking solution. Secondary antibody incubations were performed for 1 h at room temperature in a blocking solution. After washing three times, bands were visualized using Clarity Western ECL substrate (Bio-Rad, Hercules, CA, USA, #1705061) and calibrated by the Chemidoc Imaging System (Bio-Rad, Hercules, CA, USA). Appendix A lists antibodies used in this study.

## 3. Results

### 3.1. puf-8(q725) Mutation Enhances fem-3(q20gf) Mog Phenotype

Most *puf-8(q725)* single mutants (1–2 days old adult) are similar to *wild-type* worms at permissive temperatures (15–20 °C) [21,22,23,24,37]. However, at the restrictive temperature (~25 °C), *puf-8(q725)* mutant males have significantly more spermatocyte dedifferentiation-mediated germline tumors than hermaphrodites [23,24]. This finding led us to test whether excess sperm production in *puf-8(q725)* hermaphrodite germlines could enhance spermatocyte dedifferentiation. To this end, we employed a temperature-sensitive *fem-3(q20)* gain-of-function (gf) mutant (henceforth called *fem-3(q20gf)*). The *C. elegans fem-3* gene is required for spermatogenesis in both hermaphrodite and male germlines [38]. Thus, *fem-3(q20gf)* mutants produce only sperm without switching into oogenesis, even in hermaphrodite germlines at 25 °C (Figure 2A). *puf-8(q725)*; *fem-3(q20gf)* double mutants were generated by a standard genetic process (see Materials and Methods). The germline phenotypes of *puf-8(q725)* single, *fem-3(q20gf)* single, and *puf-8(q725)*; *fem-3(q20gf)* double mutants were determined by staining dissected gonads with anti-MSP antibodies (a marker for sperm fate cells) [35,39] and DAPI (a marker for DNA). Most *puf-8(q725)* and *fem-3(q20gf)* single mutants produce both sperm and oocytes, and they are self-fertile at the permissive temperature (15–20 °C), but *puf-8(q725)*; *fem-3(q20gf)* double mutants produce sperm continuously throughout adulthood (masculinization of the germline (Mog) phenotype) at the permissive temperatures (15–20 °C) (Figure 2A) such as *puf-8(q725)*; *lip-1(zh15)* mutants (Figure 1F). To confirm this phenotype, we generated a *lin-41(tn1541[GFP::tev::s::lin-41])*; *puf-8(q725)*; *fem-3(q20gf)* mutant. The *lin-41(tn1541[GFP::tev::s::lin-41])* allele was used to visualize oogenic cells (Figure 2B,C) [40]. Most *wild-type(N2)* and *puf-8(q725)* mutant worms produced sperm (MSP-positive) and oocytes (GFP::LIN-41-positive) at 15, 20, and 25 °C (Figure 2B). However, most *puf-8(q725)*; *fem-3(q20gf)* hermaphrodite mutants produced only sperm without switching to oogenesis at 15–20 °C (MSP-positive and GFP::LIN-41-negative) (Figure 2C). This result suggests that the *puf-8(q725)* mutation enhances the *fem-3(q20gf)* Mog phenotype.

### 3.2. RNAi of fog-1, fog-2, or fog-3 Rescues puf-8(q725); fem-3(q20gf) Mog Phenotype

In both hermaphrodites and males, sperm production requires *fog-1*, *fog-3*, and the three *fem* genes [41] (Figure 3A). Mutations in any of these genes cause all germ cells to differentiate as oocytes, the Fog (feminization of the germline) phenotype, and mutations in *tra* genes cause hermaphrodites to make sperm instead of oocytes, or the Mog (masculinization of germline) phenotype [9,41,42,43]. Moreover, the phosphorylation state of *fog-3*, probably by MPK-1, modulates the initiation and maintenance of the *C. elegans* sperm fate program [44]. A previous epistasis study showed that *puf-8* and *fbf-1* (*C. elegans* PUF protein family) are upstream of *fog-2*, a gene thought to be near the top of the germline sex determination pathway [37]. Since FBF-1 represses the expression of *fem-3* mRNA [45], we also performed epistasis experiments. Specifically, genes required for sperm production were depleted by RNAi in *lin-41(tn1541[GFP::tev::s::lin-41])*; *puf-8(q725)*; *fem-3(q20gf)* mutants at 20 °C. Our results showed that *puf-8(q725)*; *fem-3(q20gf)* Mog phenotypes were completely suppressed by the depletion of *fog-1*, *2*, *3*, or *fem-3* (Figure 3B,C). Interestingly, RNAi of *fog-1*, *fog-2*, or *fog-3* dramatically rescued the Mog sterile phenotype and made them fertile (56.4 ± 8.9 %, *n* = 120). These results indicate that the *puf-8* gene inhibits sperm fate at the top of the germline sex determination pathway.

### 3.3. MPK-1 Dependence of puf-8(q725); lip-1(zh15) and puf-8(q725); fem-3(q20gf) Mog Phenotype

The *C. elegans mpk-1* gene encodes two major transcripts, *mpk-1a* and *mpk-1b*, which produce MPK-1A and MPK-1B proteins, respectively [34,46]. The *mpk-1a* mRNA is contained entirely within *mpk-1b*, but *mpk-1b* harbors a unique exon [34]. The *mpk-1a* isoform is predominantly expressed in somatic cells, but the *mpk-1b* isoform is abundantly expressed in germ cells [34,46]. The germline-specific MPK-1B isoform promotes germline differentiation but has no apparent role in GSC proliferation. However, the soma-specific MPK-1A isoform promotes GSC proliferation non-cell autonomously [47]. We previously reported that MPK-1B activity is required for the *puf-8(q725)*; *lip-1(zh15)* Mog phenotype [25]. Either *mpk-1b* RNAi or germline-specific *mpk-1(ga111)* mutation dramatically rescued *puf-8(q725)*; *lip-1(zh15)* Mog sterility [25] (Figure 4A). To test whether *puf-8(q725)*; *fem-3(q20gf)* Mog sterility is also dependent on MPK-1 activity (Figure 4B), we performed *mpk-1b* RNAi in *puf-8(q725)*; *fem-3(q20gf)* mutants and determined their germline phenotype by staining dissected gonads with the anti-MSP antibody and DAPI. As previously reported [25], *mpk-1b* RNAi completely rescued the *puf-8(q725)*; *lip-1(zh15)* Mog phenotype at 20 °C (Figure 4C,E), but it partially rescued the *puf-8(q725)*; *fem-3(q20gf)* Mog phenotypes (Figure 4D,F). These results indicate that the *puf-8(q725)*; *fem-3(q20gf)* Mog phenotype depends less on the germline MPK-1B activity.

PUF-8 has been known to repress the expression of *let-60* (a Ras homolog), an upstream activator of MPK-1 [48]. MPK-1 activity is absolutely required for *puf-8(q725); lip-1(zh15)* Mog phenotypes, but not for *puf-8(q725)*; *fem-3(q20gf)* Mog phenotypes at 20 °C (Figure 4A–F) [25,26]. To ask whether *puf-8(q725)*; *lip-1(zh15)* mutant germlines have more activated (phosphorylated) MPK-1 (pMPK-1) proteins than *puf-8(q725)*; *fem-3(q20gf)* mutant germlines, we stained dissected gonads (4 days past L1) using an anti-DP-MAPK monoclonal antibody, which recognizes the active form of MPK-1 by its dual phosphorylation (DP) [34,46]. In the *puf-8(q725)*; *fem-3(q20gf)* Mog germlines, pMPK-1 proteins were detected in the early differentiating cells (Figure 4G) as seen in wild-type male germlines [46]. Notably, pMPK-1 proteins were more in the *puf-8(q725)*; *lip-1(zh15)* mutant germlines than in the *puf-8(q725)*; *fem-3(q20gf)* mutant germlines (Figure 4G,H). We also quantitated levels with ImageJ software. The average intensity of pMPK-1 in the *puf-8(q725)*; *lip-1(zh15)* mutant germlines (*n*= 16) was ~2.7 times higher than that in the *puf-8(q725)*; *fem-3(q20gf)* mutant germlines (*n* = 15) (Figure 4I). These results indicate that *puf-8(q725)*; *lip-1(zh15)* mutant germlines have a high level of pMPK-1 proteins compared to *puf-8(q725)*; *fem-3(q29gf)* mutant germlines.

### 3.4. Competence for Spermatocyte Dedifferentiation in the Absence of PUF-8

We have previously reported that excess sperm does not necessarily lead to spermatocyte dedifferentiation [23]. Rather, the activation of MPK-1 in the *puf-8 (q725)* mutant may be critical for initiating spermatocyte dedifferentiation in the *C. elegans* germline [23]. Based on this finding, we tested the hypothesis that competence for spermatocyte dedifferentiation may require MPK-1 activation in the absence of PUF-8. To this end, we also employed *wild-type(N2)* and other mutant strains—*fem-3(q20gf)* mutant with both wild-type *puf-8(+/+)* and *lip-1(+/+)* genes, *puf-8(q725)*; *fem-3(q20gf)* with the wild-type *lip-1(+/+)* gene, and *puf-8(q725)*; *lip-1(zh15)* mutants. To score the percentage of worms with germline tumors via spermatocyte dedifferentiation, synchronized L1-staged mutants were placed on NGM plates seeded with OP50 *E. coli* bacteria food at 25 °C, and their germline phenotypes were determined daily (2.5, 3–6 days past L1) by staining dissected gonads with DAPI. Although the *wild-type(N2)* and *fem-3(q20gf)* mutant worms never formed germline tumors throughout adulthood at 25 °C (Figure 5A), most *puf-8(q725)*; *lip-1(zh15)* mutants developed germline tumors even 3 days past L1 (Figure 5A). Notably, the percentage of *puf-8(q725)*; *fem-3(q20gf)* hermaphrodite mutants with germline tumors increased gradually from 3 days past L1 (Figure 5A). Since *puf-8(q725)*; *lip-1(zh15)* germline phenotypes largely depend on MPK-1 activity [25] (Figure 4), we suggest that MPK-1 activity may be required to induce the formation of germline tumors via spermatocyte dedifferentiation in the absence of PUF-8 at 25 °C.

### 3.5. Resveratrol Induces Germline Tumors by Activating MPK-1 in the Absence of PUF-8

Our previous study found that Resveratrol (RSV) maintains MPK-1 activity throughout the lifespan of *C. elegans* [49] (Figure 5B). To better understand the effect of RSV on ERK1/2, two human cell lines, MDA MB231 and WPMY-1, were treated with the different concentrations (0, 10, 20, 50 µM) of RSV for 24 h after 80% confluence. The expression levels of total ERK1/2 and phospho-ERK1/2 (pERK1/2) proteins were examined by Western blot using anti-ERK1/2 and anti-pERK1/2 antibodies (Figure 5C). RSV significantly increased the levels of pERK1/2 proteins dose-dependently up to 20 µM (Figure 5C). However, their levels significantly decreased at 50 µM RSV in both cell lines (Figure 5C). Similarly, *C. elegans* pMPK-1 levels were increased up to 3.3-fold by 100 µM RSV treatment, but their increasing levels were decreased up to 2.3-fold by 200 µM RSV treatment [49]. We also found that 50 µM RSV treatment significantly decreased cell viability in a dose-dependent manner (data not shown), as previously reported [50,51,52]. Thus, the reduction in pERK1/2 and pMPK levels in cells and worms exposed to high RSV concentration might be due to an increased cell death. This result indicates that the effects of RSV on the activation of MPK-1 and ERK1/2 are conserved in human cell lines and *C. elegans*.

Based on these findings, we tested whether 100 µM RSV could induce the formation of germline tumors via spermatocyte dedifferentiation by activating MPK-1 signaling in *puf-8(q725)*; *fem-3(q20gf)* mutant germlines. Synchronized L1 staged *puf-8(q725)*; *fem-3(q20gf)* mutant worms were cultured on NGM agar plates containing 100 µM RSV or 0.1% ethanol (EtOH) control at 25 °C. Their germline phenotypes were determined daily by staining dissected gonads with an EdU-labeling kit and DAPI. Notably, RSV significantly induced the formation of germline tumors via spermatocyte dedifferentiation from day 3 in *puf-8(q725)*; *fem-3(q20gf)* mutant germlines (Figure 5D,E). This result suggests that RSV is a potential inducer of spermatocyte dedifferentiation in vivo in the absence of PUF-8. Next, to test whether RSV-induced spermatocyte dedifferentiation relies on MPK-1 activity, we depleted the expression of *mpk-1* by RNAi from L1 staged *puf-8(q725)*; *fem-3(q20gf)* mutants in the presence of 100 µM RSV. While *vector* RNAi control did not suppress the formation of germline tumors via spermatocyte dedifferentiation, *mpk-1* RNAi significantly suppressed the formation of *puf-8(q725)*; *fem-3(q20gf)* germline tumors even in the presence of 100 µM RSV (Figure 5F). This result was confirmed by staining dissected gonads with anti-HIM-3 antibodies which recognize meiotic differentiating cells (Figure 5G). *mpk-1* RNAi inhibited the formation of germline tumors via spermatocyte dedifferentiation and induced HIM-3-postive meiotic cells (non-proliferative cells) in the *puf-8(q725)*; *fem-3(q20gf)* mutant germline (Figure 5G). Therefore, we suggest that MPK-1 activation could chemically induce the formation of germline tumors via spermatocyte dedifferentiation in the absence of PUF-8 in vivo (Figure 6).

## 4. Discussion

RNA-binding proteins (RBPs) bind to either single-stranded or double-stranded RNA and play a role in the post-transcriptional control of RNAs, such as mRNA stabilization, localization, splicing, polyadenylation, and translation [53]. PUF family RBPs are highly conserved among most eukaryotic organisms [7]. PUF proteins are conserved RBPs that maintain GSCs in worms and flies and have also been implicated in this role in mammals [12,15,17,54,55,56,57]. PUF proteins bind specifically to PUF binding elements (PBE: UGUAnAUA) within the 3′ untranslated region (3′UTR) of their direct target mRNAs to repress their translation [7,19,20] (Figure 1C). PUF proteins also have diverse roles depending on the organism. For example, in *Drosophila melanogaster*, Pumilio is required for embryonic development through the regulation of Hunchback (necessary for the establishment of an anterior-posterior gradient) [58] and GSC maintenance [59]. The yeast PUF protein Mpt5, a broad RNA regulator in *Saccharomyces cerevisiae*, binds to more than 1,000 RNA targets [60]. The Mpt5 is required to promote G2/M cell cycle progress [61] and cell wall integrity [62]. Humans have two PUF proteins, PUM1 and PUM2. The PUM1 and PUM2 have high structural similarity and recognize the same RNA binding motif. Despite their similarities, PUM1 is critical for stem cell proliferation, and PUM2 is more important for stem cell differentiation and cell lineage specification [12]. Notably, PUF proteins repress mRNAs encoding MAPK enzymes in worms, flies, yeast, and humans [34,63,64]. However, the role of PUF proteins in limiting dedifferentiation remain poorly understood.

Cellular dedifferentiation counteracts the decline of stem cells during aging but has also been implicated in the formation of tumor-initiating cells [65]. Thus, a comprehensive examination of what causes stem cells to differentiate into desired cell types and how committed cells return to undifferentiated cells is a central question in stem cell biology, regenerative medicine, and tumorigenesis [66]. This cellular dedifferentiation can take many forms depending on the specific organism and tissue type. In zebrafish (*Danio rerio*), cellular dedifferentiation occurs in a controlled environment where cardiomyocytes partially dedifferentiate to repopulate lost ventricular tissue [67]. In fruit flies (*Drosophila melanogaster*), it has been shown that differentiating germ cells can revert into functional stem cells both in second instar larval ovaries and in adult fruit flies [68]. *Drosophila* has also been shown to induce dedifferentiation in spermatogonia cells as there is considerable plasticity due to Jak-STAT signaling [69]. Reduction in stem cell division in *Drosophila* has been shown due to an accumulation of GSCs with misoriented centrosomes that increases as the flies age [70]. In *Mus musculus*, dedifferentiated basal-like cells originating from luminal airway cells can function as stem cells in the repopulation of damaged airway epithelia [71]. Dedifferentiation in the mouse model has also been shown in the intestine, where tumorigenesis is initiated due to increased Wnt-activation allowing for polyp formation [72]. In these different examples of dedifferentiation, other signaling pathways are reverting differentiated cells into either stem cell-like or tumor-initiating cells.

In *C. elegans*, dedifferentiation in the germline can occur in oogenic [73] and spermatogenic [21,24] germlines. In a mutant lacking *gld-1* (a KH-motif containing RNA-binding protein), germ cells destined for oogenesis early in the meiotic cell fate return to a mitotic cell cycle, which results in germline tumors [73]. In a double mutant lacking *puf-8* and *lip-1*, spermatocytes do not undergo normal meiotic division but instead return to mitosis, resulting in germline tumors [21,23,24] (Figure 6). Notably, PUF-8 and GLD-1 inhibit MPK-1 signaling pathways at post-transcriptional levels [48,74,75,76,77], and the formation of germline tumors via dedifferentiation was significantly inhibited by the depletion of *mpk-1* [21,23]. This result indicates that MPK-1 activity may be critical for the formation of germline tumors via dedifferentiation in the absence of PUF-8 (Figure 6) or GLD-1. Consistently, mammalian Ras-ERK MAPK signaling has similarly been implicated in the cellular dedifferentiation of the Sertoli cells [78], myoblasts [79], and islet cells [80]. In addition to MPK-1, our results using *puf-8(q725)*; *fem-3(q20gf)* and *fem-3(q20gf)* mutants demonstrated that spermatocyte dedifferentiation also requires *puf-8* loss. We also examined the germline phenotypes of other PUF mutants in the absence of LIP-1, such as *fbf-1(ok91)*; *lip-1(zh15)* [23], *fbf-2(q704)*; *lip-1(zh15)* [23], and *puf-9(ok1136)*; *lip-1(zh15)* (Jones et al., unpublished result). None of them formed germline tumors, indicating that PUF-8 has a special function in inhibiting spermatocyte dedifferentiation. Therefore, the identification and characterization of PUF-8 target genes involved in spermatocyte dedifferentiation will be critical to understanding the mechanism of spermatocyte dedifferentiation.

During the testing of germline tumors with *mpk-1* RNAi, we found that *puf-8(q725)*; *lip-1(zh15)* mutants fed OP50 had a lower percentage of tumor formation than that fed HT115, an RNase III-deficient *E. coli* strain used for feeding RNAi in *C. elegans* [81,82] (see Figure 5D,F; Appendix A). The HT115 *E. coli* provided a greater metabolic energy source due to the recycling of excess nucleotides by the bacteria being RNase III-deficient [83]. Due to this increase in energy boosting the formation of germline tumors, we decided to look at the levels of active MPK-1. Notably, the worms fed HT115 *E. coli* had increased pMPK-1 than those fed OP50 *E. coli* (unpublished result). These unpublished results suggest that the HT115 diet may increase spermatocyte dedifferentiation via activating MPK-1 signaling. Notably, increasing evidence suggests that metabolic changes alter cell fates by changing multiple signaling pathways. For example, starvation or starvation-induced quiescence maintains GSCs, independent of GLP-1/Notch signaling [84,85]. Furthermore, short-term starvation stress enhances the meiotic activity of germ cells to prevent age-related declines in sperm production [86]. Similarly, the dedifferentiation of primary hepatocytes is accompanied by the reorganization of lipid metabolism [87]. Therefore, non-genetic factors may play a vital role in cell fate reprogramming and tumorigenesis. Our findings reveal fundamental mechanisms of the differentiation/dedifferentiation decision in vivo and may provide a future platform for identifying therapeutic targets for dedifferentiation-mediated tumorigenesis.

## Figures and Tables

**Figure 1 cells-12-00434-f001:**
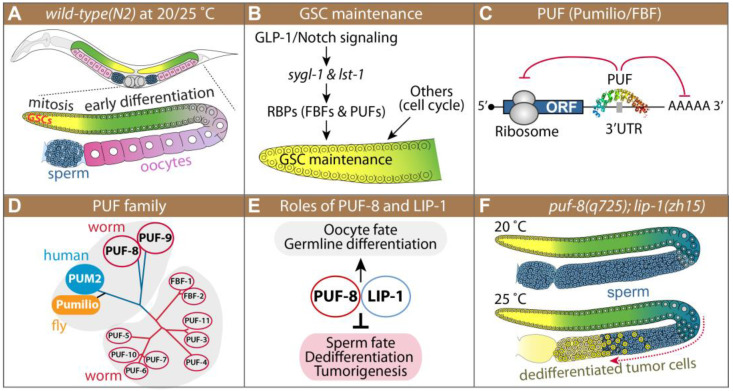
Introduction. (**A**) Schematics of adult *C. elegans* and its germline. Germ cells at the distal end of the germline, including GSCs, divide mitotically (yellow). As germ cells move proximally, they enter meiosis (green) and differentiate into either sperm (blue) or oocytes (pink). (**B**) GLP-1/Notch signaling and intrinsic regulators control GSC maintenance. (**C**) PUF as a translational repressor. (**D**) The PUF protein family is widespread throughout eukaryotes. The phylogenetic tree is adapted from Wickens’ 2002 review [7]. (**E**) PUF-8 and LIP-1 control many cellular processes. (**F**) Schematics of *puf-8(q725)*; *lip-1(zh15)* germline phenotypes at 20 °C and 25 °C. Yellow circles in proximal gonads indicate dedifferentiated tumor cells.

**Figure 2 cells-12-00434-f002:**
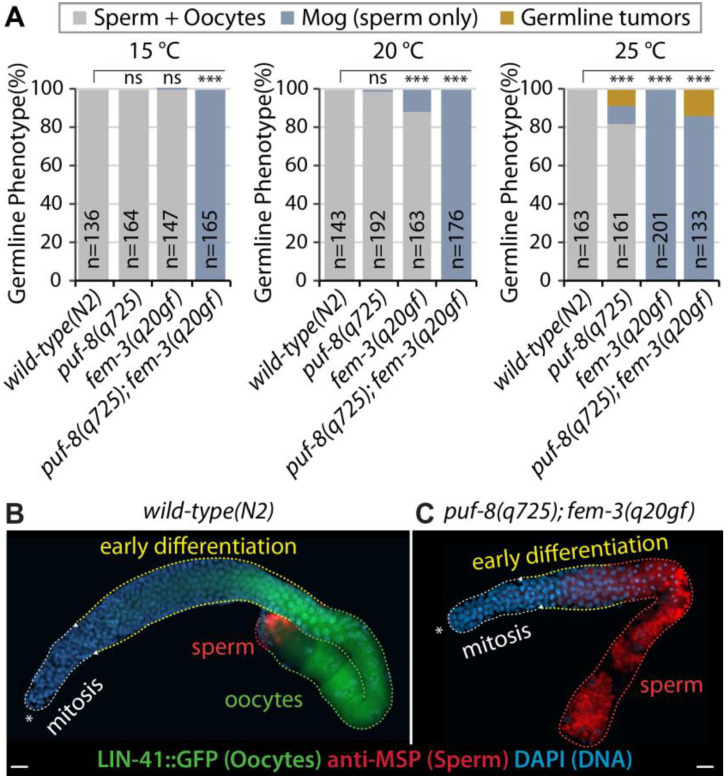
Phenotype analysis of *puf-8(q725)*; *fem-3(q20gf)* germlines. (**A**) Germline phenotypes were determined by staining dissected gonads with anti-MSP antibodies and DAPI. (**B**,**C**) Expression of LIN-41::GFP (oocyte marker) and MSP (sperm marker) in *wild-type(N2)* and *puf-8(q725)*; *fem-3(q20gf)* mutant germlines. Scale bars are 10 μm. All experiments were performed in triplicate. The statistical significance was assessed by Student’s t-test without corrections for multiple comparisons. ***, *p* < 0.001; ns, not statistically significant.

**Figure 3 cells-12-00434-f003:**
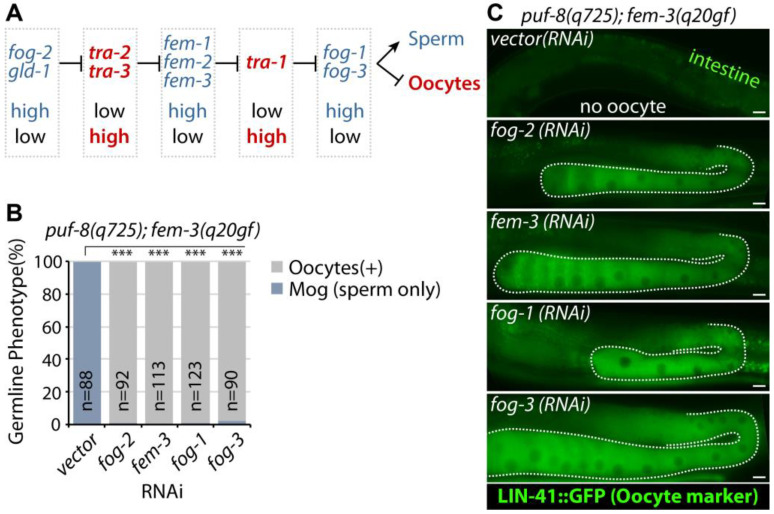
Epistatic analysis. (**A**) A simplified version of the germline sex determination pathway. Red genes promote oocyte fate, and blue genes promote sperm fate. (**B**) Depletion of sperm-promoting genes by RNAi rescues *puf-8(q725)*; *fem-3(q20gf)* Mog sterility. (**C**) The expression of LIN-41::GFP (oocyte marker). Scale bars are 10 μm. All experiments were performed in triplicate. The statistical significance was assessed by Student’s t-test without corrections for multiple comparisons. ***, *p* < 0.001.

**Figure 4 cells-12-00434-f004:**
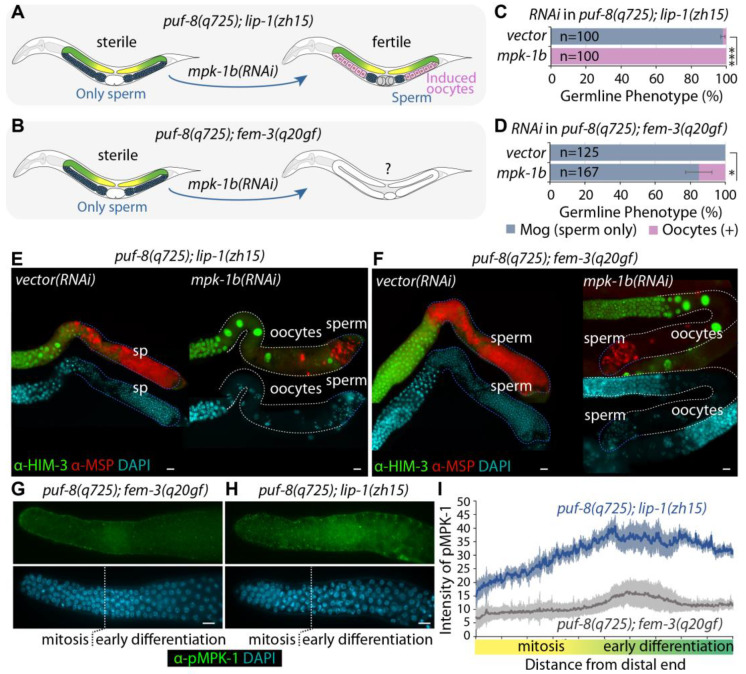
MPK-1B dependence of *puf-8(q725)*; *lip-1(zh15)* and *puf-8(q725)*; *fem-3(q20gf)* Mog phenotypes. (**A**) MPK-1B dependence of *puf-8(q725)*; *lip-1(zh15)* Mog sterility. (**B**) The potential role of *mpk-1b* in *puf-8(q725)*; *fem-3(q20gf)* Mog sterility. (**C**,**D**) *mpk-1b* RNAi completely rescues *puf-8(q725)*; *lip-1(zh15)* Mog sterility, but partially rescues *puf-8(q725)*; *fem-3(q20gf)* Mog sterility. All experiments were performed in triplicate. The statistical significance was assessed by Student’s t-test without corrections for multiple comparisons. ***, *p* < 0.001; *, *p* < 0.05. (**E**,**F**) Germline staining with anti-HIM-3 and anti-MSP antibodies. Scale bars are 10 μm. (**G**,**H**) Germline staining with anti-DP-MAPK monoclonal antibody and DAPI. Germlines (4 days past L1) were treated identically, and images were taken with the same settings at the same magnification for comparison. Scale bars are 10 μm. (**I**) Quantitation of pMPK-1 proteins in *puf-8(q725)*; *fem-3(q20gf)* (*n* = 15) and *puf-8(q725)*; *lip-1(zh15)* (*n* = 15) mutant germlines. The intensity of pMPK-1 proteins was quantified using ImageJ software. The *x*-axis represents the distance from the distal end of the germline, and the *y*-axis is pixel intensity. Error bars are standard deviations.

**Figure 5 cells-12-00434-f005:**
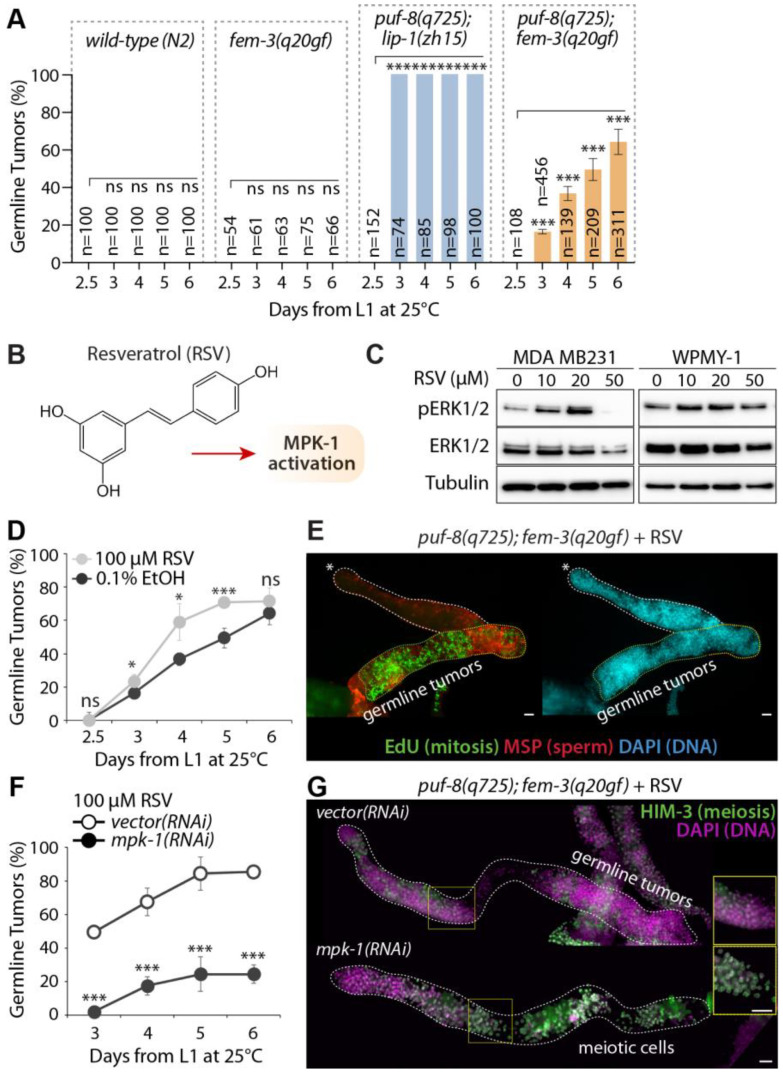
The activation of MPK-1 induces the formation of germline tumors via spermatocyte dedifferentiation in the absence of PUF-8. (**A**) % germline tumors. (**B**) Chemical structure of Resveratrol and its effect on MPK-1 activation. (**C**) Western blot. (**D**) The percentage of germline tumors at 25 °C. The germline phenotypes were determined at 2.5, 3, 4, 5, and 6 days after the L1 stage. (**E**) Staining of dissected adult hermaphrodite germline with EdU-labeling kit, anti-MSP, and DAPI. Scale bars are 10 μm. (**F**) The percentage of germline tumors at 25 °C. The germline phenotypes were determined at 2.5, 3, 4, 5, and 6 days after the L1 stage. (**G**) Staining of dissected adult hermaphrodite germline with anti-HIM-3 and DAPI. Scale bars are 10 μm. All experiments were performed in triplicate. The statistical significance was assessed by Student’s t-test without corrections for multiple comparisons. ***, *p* < 0.001; *, *p* < 0.05; ns, not statistically significant.

**Figure 6 cells-12-00434-f006:**
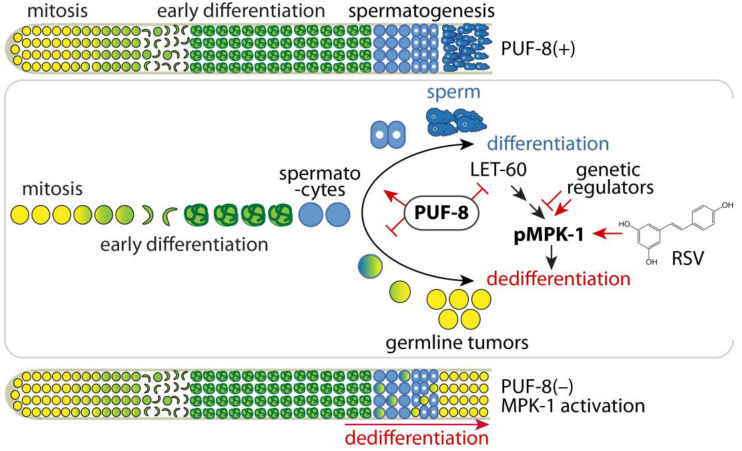
A model for the spermatocyte differentiation/dedifferentiation decision. Top: The schematic of normal spermatogenesis in the *C. elegans* germline. Middle: Genetic and chemical regulation of the spermatocyte differentiation/dedifferentiation decision. Bottom: Schematic of spermatocyte dedifferentiation-mediated tumorigenesis via PUF-8 loss and activation of MPK-1 signaling.

## Data Availability

Not applicable.

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
