# Peer review of "Genetic and Chemical Controls of Sperm Fate and Spermatocyte Dedifferentiation via PUF-8 and MPK-1 in Caenorhabditis elegans"

_cells, 2023, doi:10.3390/cells12030434_

Round 1

Reviewer 1 Report

How committed cells return to undifferentiated cells through cellular dedifferentiation is an intriguing question in stem cell biology. PUF proteins are a conserved family of RNA binding proteins that have been implicated in cell proliferation or cell linages specification however their role in dedifferentiation remain poorly understood. In this manuscript, Park et al. report that the excess of sperm does not lead cells to dedifferentiation but instead the MPK-1 (the C. elegans ERK1/2 MAPK ortholog) is necessary for this phenomenon. This result indicates that MPK-1 activity may be critical for the formation of germline tumors via dedifferentiation. The authors further prove that MPK-1 activation by exposing animals to resveratrol promotes dedifferentiation. The manuscript is well written and the experiments are well designed and performed. The role of MPK-1 in cell dedifferentiation has been described in other organisms however the role of pumilo in this process is novel and remains a mystery. This work provides strong evidence for future studies that could be done to elucidate the mechanism by which PUF-8 inhibits cell dedifferentiation while MPK-1 promotes it.

Major corrections:

1. This reviewer found difficult to understand the contribution of this manuscript versus the previous work (BBA, 2012). Please provide a clearer explanation between the findings and novelty of each work. Perhaps an explanation directed to a broader audience instead to C. elegans experts will be helpful.

2. From reading the manuscript is not clear whether mpk-1 mRNA is target of PUF-8 regulation. Does it have PUF-8 binding sites?

Minor corrections:

-Line 68 please change C. elegans have for…….C. elegans has.

-Please provided which statistic test was used in each figure.

-Line 407 described the roles of Mtp-5 in Drosophila.

-Does any of the Mtp-5 targets belong to genes that enconde for the mapk familiy?

Author Response

Response to Reviewer #1’s Comments

We thank the reviewer for her/his time and effort in reviewing our manuscript and for her/his invaluable insights that undoubtedly improved this work. Kindly find below our point-by-point responses to each comment verbatim.

Major corrections:

1. This reviewer found difficult to understand the contribution of this manuscript versus the previous work (BBA, 2012). Please provide a clearer explanation between the findings and novelty of each work. Perhaps an explanation directed to a broader audience instead to C. elegans experts will be helpful.

--> We appreciate the reviewer’s helpful comment. Previous works reported that PUF-8 and LIP-1 normally repress spermatocyte dedifferentiation by inhibiting MPK-1/ERK MAPK signaling. Current works were focused on the competence for spermatocyte dedifferentiation using three mutants  – fem-3(gf), puf-8; fem-3(gf), and puf-8; lip-1. To this end, we carefully characterized the germline phenotypes of puf-8; fem-3(gf) as a counterpart worm for spermatocyte dedifferentiation (Figures 2 and 3). Our finding was that puf-8; lip-1 mutants with high pMPK-1 levels had more germline tumor formation via dedifferentiation than puf-8; fem-3(gf) mutants with less pMPK-1 levels (see Figure 5). We also found that activated MPK-1 proteins themselves failed to induce germline tumor formation in the presence of PUF-8 (see Figure 5). These results indicate that competence for dedifferentiation correlates with PUF-8 loss and high MPK-1 activity. This finding was confirmed using a potential activator (resveratrol) of MPK-1/ERK in the absence of PUF-8 (see Figure 6). We also found for the first time that different bacteria diets (HT115) can induce spermatocyte dedifferentiation in the absence of PUF-8 (Figure S2 and p501-519). The importance of our findings was also described in the discussion section (lines 460-478).  

2. From reading the manuscript is not clear whether mpk-1 mRNA is target of PUF-8 regulation. Does it have PUF-8 binding sites?

--> We appreciate the reviewer’s comment. PUF-8 physically interacts with the 3’UTR of let-60 (a Ras homolog), an upstream activator of MPK-1, and represses its expression [48]. Our in silico analysis did not nominate mpk-1 mRNA for a putative target [21]. Thus, we believe that PUF-8 represses the activation of MPK-1 by inhibiting the expression of let-60/Ras (see lines 291-292 and Figure 6). 

Minor corrections:

-Line 68 please change C. elegans have for…….C. elegans has.

--> We changed it.

-Please provided which statistic test was used in each figure.

--> We added it.

-Line 407 described the roles of Mtp-5 in Drosophila

--> We described it (see lines 452).

-Does any of the Mtp-5 targets belong to genes that enconde for the mapk familiy?

--> Yes, we added this information and reference to the revised version (see lines 456-457).

Reviewer 2 Report

In this manuscript, Park and colleagues investigate spermatocyte de-differentiation observed in several masculinized genetic backgrounds of the nematode C. elegans. The authors find that de-differentiation is dependent on the activity of MAP kinase homolog mpk-1, and can be potentiated by a treatment with resveratrol. The authors further report complex dose-dependent effects of resveratrol on pMAPK in human tissue culture cells. This is a thorough report that provides important insight in cell fate stability vs de-differentiation. However, there are several issues that need to be addressed prior to publication: 

Major items:

1. Lines 26-28, 74, 90, 441, 342 (title of Fig. 5) and Figure 6 need to be revised to reflect the ambiguity of the connection between lip-1 and pMPK-1. The authors’ claim that lip-1(zh15) mutation activates MAP kinase in C. elegans germline has recently been challenged (Das et al., 2021), and this needs to be reflected in the text. The motivation of the experiments could be changed to test the role of mpk-1 in de-differentiation without claiming that MPK-1 activation is the result of lip-1 mutation.

2. Figure 2A: Did this experiment involve one or several (how many?) replicates? What were the statistical tests used to analyze the data? Were all mutants compared to the wild type? Were there any corrections for multiple comparisons?

3. Figure 3B: Indicate the number of replicates done for this experiment. What were the statistical tests used to analyze the data? Were there any corrections for multiple comparisons? 

4. Figure 5A: How many germlines were scored? Did this experiment involve one or several (how many?) replicates? What was compared for significance (%tumors compared to d2.5 in each strain? Or % tumors for each day is a strain compared to all other days?)?  What was the statistical test used to determine significant difference, and was there a correction for multiple comparisons? Why is there a ns bar connecting d5 and d6 of puf-8, should we assume that the other values are different from each other as well as from d2.5? – surely, this can’t be true for puf-8; lip-1 as it remains at 100% for 4 days?

Minor suggestions:

1. Figure 1B: the schematic makes it look like sygl-1 and lst-1 function in parallel to FBFs, which isn’t the current model (Haupt et al., 2020). The challenge may be to effectively communicate that PUF-8 might function in parallel to sygl-1 and lst-1.

2. Line 62 and Figure 1D: If this panel is adapted from the Wickens’ 2002 review, the reference should be provided in the legend. Otherwise, please describe how the phylogenetic tree was constructed (eg, based just on the PUF domains or on the whole protein sequences).

3. Section 2.4; lines 229, 258; Table S1: The tn1541 allele is on LG I, therefore all triple allele strains need to be appropriately labeled as lin-41(tn1541[GFP::tev::s::lin-41]; puf-8(q725); fem-3(q20gf)

4. Line 329: past, not “paster”

5. Figure 5D: the symbols in the legend and the graph do not match.

6. Figure 5: since Fig. 5A shows the dynamics of tumor formation are very similar in the single puf-8 vs. puf-8; fem-3 mutants, I was wondering whether treatment of puf-8(q725) with RSV similarly promoted dedifferentiation? [Note: I don’t think this experiment is necessary for the paper, but if the authors have relevant data, it would help better understand cell signaling requirements for dedifferentiation in different genetic backgrounds].

7. Line 413: it would make more sense to say “the role of PUFs in limiting dedifferentiation”

Author Response

We thank the reviewer for her/his time and effort in reviewing our manuscript and for her/his invaluable insights that undoubtedly improved this work. Kindly find below our point-by-point responses to each comment verbatim.

Major Items

1. Lines 26-28, 74, 90, 441, 342 (title of Fig. 5) and Figure 6 need to be revised to reflect the ambiguity of the connection between lip-1and pMPK-1. The authors’ claim that lip-1(zh15)mutation activates MAP kinase in C. elegans germline has recently been challenged (Das et al., 2021), and this needs to be reflected in the text. The motivation of the experiments could be changed to test the role of mpk-1 in de-differentiation without claiming that MPK-1 activation is the result of lip-1 mutation.

-->  We appreciate the reviewer’s helpful comments. We revised our manuscript and figures appropriately as per the reviewer’s direction and added Figures 4G-4I to the revised version.

Although Das et al. reported that LIP-1 does not function as an MPK-1 DUSP in the C. elegans germline, our previous Nature Chemical Biology paper showed that the inhibition of MPK-1 activity either genetically (mpk-1b RNAi or mpk-1(ga111) mutation) or chemically (U0126 (a MEK chemical inhibitor) treatment) comp koely rescued puf-8; lip-1 Mog phenotype [25].  This result strongly suggests that LIP-1 inhibits MPK-1 signaling, at least in puf-8 mutant germlines. To confirm this result, we here compared the levels of pMPK-1 between puf-8; fem-3(gf) and puf-8; lip-1 mutant germlines using DP-MAPK(YT) monoclonal antibodies. Notably, pMPK-1 levels were higher in puf-8; lip-1 than in puf-8; fem-3(gf) mutant germlines (see new Figures 4G-4I in the revised version and lines 291-306).

2. Figure 2A: Did this experiment involve one or several (how many?) replicates? What were the statistical tests used to analyze the data? Were all mutants compared to the wild type? Were there any corrections for multiple comparisons?

--> We added the statistical methods to the Figure 2 legend.

3. Figure 3B: Indicate the number of replicates done for this experiment. What were the statistical tests used to analyze the data? Were there any corrections for multiple comparisons? 

--> We added the statistical methods to the Figure 3 legend.

4. Figure 5A: How many germlines were scored? Did this experiment involve one or several (how many?) replicates? What was compared for significance (%tumors compared to d2.5 in each strain? Or % tumors for each day is a strain compared to all other days?)?  What was the statistical test used to determine significant difference, and was there a correction for multiple comparisons? Why is there a ns bar connecting d5 and d6 of puf-8, should we assume that the other values are different from each other as well as from d2.5? – surely, this can’t be true for puf-8; lip-1 as it remains at 100% for 4 days?

--> We added the statistical methods to the Figure 5 legend and corrected the error bars in the graphs.

Minor suggestions:

1. Figure 1B: the schematic makes it look like sygl-1and lst-1function in parallel to FBFs, which isn’t the current model (Haupt et al., 2020). The challenge may be to effectively communicate that PUF-8 might function in parallel to sygl-1 and lst-1.

--> We appreciate the reviewer’s helpful comment. We revised Figure 1B accordingly.

2. Line 62 and Figure 1D: If this panel is adapted from the Wickens’ 2002 review, the reference should be provided in the legend. Otherwise, please describe how the phylogenetic tree was constructed (eg, based just on the PUF domains or on the whole protein sequences).

--> We added the sentence and reference to the Figure 1E legend. 

3. Section 2.4; lines 229, 258; Table S1: The tn1541allele is on LG I, therefore all triple allele strains need to be appropriately labeled as lin-41(tn1541[GFP::tev::s::lin-41]; puf-8(q725); fem-3(q20gf)

--> We appreciate the reviewer’s helpful comment. We revised the section (see lines 138-152).

4. Line 329: past, not “paster”

--> We corrected it.

5. Figure 5D: the symbols in the legend and the graph do not match.

--> We corrected it.

6. Figure 5: since Fig. 5A shows the dynamics of tumor formation are very similar in the single puf-8vs. puf-8; fem-3mutants, I was wondering whether treatment of puf-8(q725) with RSV similarly promoted dedifferentiation? [Note: I don’t think this experiment is necessary for the paper, but if the authors have relevant data, it would help better understand cell signaling requirements for dedifferentiation in different genetic backgrounds].

--> I agree with the reviewer’s comment. The puf-8(q725) graph in Figure 5A was deleted in the revised version. 

7. Line 413: it would make more sense to say “the role of PUFs in limiting dedifferentiation”

--> We changed it (see line 458).

Reviewer 3 Report

Review for the manuscript entitled "Genetic and Chemical Controls of Sperm Fate and Spermatocyte Dedifferentiation in Caenorhabditis elegans" by Youngyong Park et al.  [Paper #  cells-2140068] 

The manuscript by Park et al. addressed the role of PUF-8 and MPK-1 in spermatocyte dedifferentiation and tumorigenesis in the germline of C. elegans. They first showed the phenotype of puf-8(q725); fem-3(q20gf) that produces excess sperm with a low prevalence of germline tumors at the restrictive temperature. They hypothesized that this phenotype is due to the de-repression of MPK-1 in the absence of PUF-8. They used Resveratrol, a potential activator of MPK-1, and found that spermatocyte dedifferentiation was enhanced by RSV treatment. Furthermore, its effects were blocked by mpk-1 RNAi. In this study, the authors showed that PUF-8 and MPK-1 are required to inhibit spermatocyte dedifferentiation and tumorigenesis. This study, however, found a lack of evidence of detailed molecular mechanisms underlying how PUF-8 inhibits MPK-1 during spermatogenesis to inhibit spermatocyte dedifferentiation. Overall, this study is interesting by providing a novel genetic pathway in the differentiation/dedifferentiation decision as well as identifying therapeutic targets for tumorigenesis. However, I would like to suggest a few changes and several important comments that would need to be addressed.

1.     The title is too broad and sounds like the title of a review article. Please revise it more precisely.

2.     The abstract should be revised. It contains too many unnecessary details and the purpose of this study is unclear.

3.     It is unclear why the authors used puf-8; fem-3(gf) rather than puf-8; lip-1 double mutant for the majority of experiments. More explanation is required.

4.     Fig 5: What effect does RSV have on puf-8 and lip-1?

5.     Fig 5: Is there any further discussion on the opposite effect of RSV based on dosage?

6.     What effects on MPK-1 by PUF-8? Is MPK-1 a direct target of PUF-8? mRNA expression in puf-8(0)? MPK-1 protein level?  pMPK-1? Dose mpk-1 overexpression cause germline tumors?

7.     Fig 1D: the letters are not visible. Please use letters in black.

8.     Line 329: paster L1 -> past L1

Author Response

We thank the reviewer for her/his time and effort in reviewing our manuscript and for her/his invaluable insights that undoubtedly improved this work. Kindly find below our point-by-point responses to each comment verbatim.

1. The title is too broad and sounds like the title of a review article. Please revise it more precisely.

--> We appreciate the reviewer’s comment. We revised the title accordingly.

2. The abstract should be revised. It contains too many unnecessary details and the purpose of this study is unclear.

--> We revised the abstract.

3. It is unclear why the authors used puf-8; fem-3(gf) rather than puf-8; lip-1 double mutant for the majority of experiments. More explanation is required.

--> We generated and characterized puf-8; fem-3(gf) as a counterpart worm for puf-8; lip-1 to identify key determinants for spermatocyte dedifferentiation. Current works found that competence for spermatocyte dedifferentiation correlates with PUF-8 loss and high MPK-1 activity. This idea was confirmed by the treatment of a potential activator, Resveratrol, in puf-8; fem-3(gf) mutant worms. The revised manuscript included this information.

4. Fig 5: What effect does RSV have on puf-8 and lip-1?

--> We found that RSV induces germline tumor formation via dedifferentiation in the puf-8 mutant, but not in the lip-1 mutant. We included this result in the original Figure 5A, but reviewer 2 recommended to delete in the revised version because the dynamics of tumor formation are very similar in the puf-8 vs puf-8; fem-3(gf) mutants.

5. Fig 5: Is there any further discussion on the opposite effect of RSV based on dosage?

--> The opposite effect of a high RSV dose on pERK1/2 levels may be due to increased cell death. This observation was added to the revised text (see line 359).

6. What effects on MPK-1 by PUF-8? Is MPK-1 a direct target of PUF-8? mRNA expression in puf-8(0)? MPK-1 protein level?  pMPK-1? Dose mpk-1 overexpression cause germline tumors?

--> We appreciate the reviewer’s comment. PUF-8 physically interacts with the 3’UTR of let-60 (a Ras homolog), an upstream activator of MPK-1, and represses its expression [48]. Our in silico analysis did not nominate mpk-1 mRNA for a putative target [21]. Thus, we believe that PUF-8 represses the activation of MPK-1 by inhibiting the expression of let-60/Ras mRNAs (see lines 291-292 and Figure 6). We also found that activated MPK-1 proteins themselves failed to induce germline tumor formation in the presence of PUF-8 (see Figure 5). These results indicate that competence for dedifferentiation correlates with PUF-8 loss and high MPK-1 activity (see Figure 6).

7. Fig 1D: the letters are not visible. Please use letters in black.

--> We changed it.

8. Line 329: paster L1 -> past L1

--> We changed it.

Reviewer 4 Report

This is a clearly written (especially given the complexity of the regulatory system in the C. elegans germline) and well documented study of the role of MAPK and PUF RNA binding proteins in maintenance of germline phenotypes. The authors provide excellent diagrams to accompany their aesthetically pleasing figures. This is a substantial study and should be of general interest to those studying germline maintenance and those interested in the role of de-differentiation in tumorigenesis.

I only have minor editorial suggestions.

23 Delete ‘our results show that’ to reduce wordiness

25 Does ‘Aggressive’ mean ‘penetrant’ or ‘severe’? Recommend replacing word for clarity.

42, 44, 56 Delete ‘The’

45 Delete ‘and they are hence self-fertile’

49, 291 Add ‘The’ before ‘C. elegans’

79-80 Could use a reference

211 Replace ‘normal, like’ with ‘similar to’

293, 295 Delete ‘It was reported that’ and ‘It was recently reported that’ because that phrase will lead the reader to expect that you will show something to the contrary.

329 ‘paster’ should be ‘post’

330 gonads

398 Edit to ‘PUF family RBPs are highly conserved among most eukaryotic organisms’. Or just delete the sentence, since it is redundant with the next one.

403 Clarify. Do they repress their translation and decrease stability?

411 Delete ‘increasing evidence showed that’

457 Write sentence in active voice, otherwise it seems like this is someone else’s finding.

460 Delete ‘It was known that’

464 Delete ‘more’

465 Edit ‘propose’ to ‘suggest’ – although depending on journal rules, it might not be appropriate to bring up unpublished results or data not shown in the discussion

467 Edit ‘has been reported’ to ‘suggests’

472 Edit ‘Their data indicate that’ to ‘Therefore’

473 Edit ‘playmaker’ to ‘role’

473… Edit to ‘Our findings reveal fundamental mechanisms of the differentiation/dedifferentiation decision in vivo and may provide a future platform for identifying therapeutic targets for dedifferentiation-mediated tumorigenesis.’

Please add scale bars to all the gonad images.

Author Response

Response to Reviewer #4’s Comments

We thank the reviewer for her/his time and effort in reviewing our manuscript and for her/his invaluable insights that undoubtedly improved this work. Kindly find below our point-by-point responses to each comment verbatim.

23 Delete ‘our results show that’ to reduce wordiness

--> We deleted it.

25 Does ‘Aggressive’ mean ‘penetrant’ or ‘severe’? Recommend replacing word for clarity.

--> We changed it.

42, 44, 56 Delete ‘The’

--> We deleted it.

45 Delete ‘and they are hence self-fertile’

--> We deleted it.

49, 291 Add ‘The’ before ‘C. elegans’

--> We added it.

79-80 Could use a reference

--> We added a reference.

211 Replace ‘normal, like’ with ‘similar to’

--> We changed it.

293, 295 Delete ‘It was reported that’ and ‘It was recently reported that’ because that phrase will lead the reader to expect that you will show something to the contrary.

--> We appreciate the reviewer’s comment. We deleted it.

329 ‘paster’ should be ‘post’

--> We changed it.

330 gonads

--> We changed it.

398 Edit to ‘PUF family RBPs are highly conserved among most eukaryotic organisms’. Or just delete the sentence, since it is redundant with the next one.

--> We deleted it.

403 Clarify. Do they repress their translation and decrease stability?

--> We revised it.

411 Delete ‘increasing evidence showed that’

--> We deleted it.

457 Write sentence in active voice, otherwise it seems like this is someone else’s finding.

--> We revised it.

460 Delete ‘It was known that’

--> We deleted it.

464 Delete ‘more’

--> We deleted it.

465 Edit ‘propose’ to ‘suggest’ – although depending on journal rules, it might not be

appropriate to bring up unpublished results or data not shown in the discussion

--> We edited it.

467 Edit ‘has been reported’ to ‘suggests’

--> We edited it.

472 Edit ‘Their data indicate that’ to ‘Therefore’

--> We edited it.

473 Edit ‘playmaker’ to ‘role’

--> We edited it.

473… Edit to ‘Our findings reveal fundamental mechanisms of the differentiation/dedifferentiation decision in vivo and may provide a future platform for identifying therapeutic targets for dedifferentiation-mediated tumorigenesis.’

--> We edited it.

Round 2

Reviewer 3 Report

The authors have addressed the majority of my concerns. I believe that this version of the manuscript is improved, and I don't have any other requirements.